# A Global Program-Educational-Objectives Comparative Study for Malaysian Electrical and Electronic Engineering Graduates

**Mohammad Syuhaimi Ab-Rahman [1,\*], I-Shyan Hwang [2], Abdul Rahman Mohd Yusoff [1], Abdul Wahab Mohamad [1], Ahmad Kamal Ariffin Mohd Ihsan [1], Juwairiyyah Abdul Rahman [3], Mohd Jailani Mohd Nor [4] and Iszan Hana Kaharudin [5]**

[1] Faculty of Engineering and Built Environment, Universiti Kebangsaan Malaysia, Bandar Baru Bangi 43600, Malaysia; abdrahman.mdyusoff@yahoo.com (A.R.M.Y.); awm.ukm@gmail.com (A.W.M.); kamal3@ukm.edu.my (A.K.A.M.I.)

[2] Department of Computer Science and Engineering, Yuan-Ze University, Chung-Li 32003, Taiwan; ishwang@saturn.yzu.edu.tw

[3] Faculty of Engineering, Universiti Selangor (UNISEL), Batang Berjuntai 45600, Malaysia; juwairiyyah@unisel.edu.my

[4] Fakulti Kejuruteraan Mekanikal, Universiti Teknikal Malaysia (UTeM), Durian Tunggal 76100, Malaysia; jai@utem.edu.my

[5] Centre for Liberal Studies (CITRA), Universiti Kebangsaan Malaysia, Bandar Baru Bangi 43600, Malaysia; iszanhana@ukm.edu.my

\* Correspondence: syuhaimi@ukm.edu.my

**Abstract:** Outstanding academic achievement in the field of higher education is a source of pride for the university. The success of the university is measured not only by academic performance but also by the quality of graduates produced. In Malaysia, three major categories in higher learning are identified: public, private, and foreign-branch universities. All engineering programs follow the requirements set by the Engineering Accreditation Council (EAC) on behalf of the Board of Engineers Malaysia (BEM). The programme educational objectives (PEOs) make up one of the elements that needs assessment for ensuring its continuity in line with the university's mission and vision. A PEO comparative study on selected reputable electrical and electronic (EE)-engineering department universities was carried out based on the mapping of PEO attribute keywords. These attributes were then classified into either cluster, sharing, or uniqueness groups. The study compared the relevancy of each PEO statement suggested by stakeholders and other interested parties. The results from the PEO comparative study suggested that attributes on competency, ethics, professionalism, and leadership are given high priorities. However, the increase in demand for entrepreneurship-, multidisciplinary-, and soft skills should also be considered when reviewing the institution's engineering curriculum. The uniqueness of such attributes will distinguish the EE-engineering graduates' profession, marketability, and employability. PEO statements reflect the credibility and sustainability of a well-balanced graduate equipped with the right knowledge, skills, and values.

**Keywords:** accreditation; programme educational objectives (PEOs); graduates' attributes; institutes of higher learning (IHL); electrical and electronic (EE); sustainable engineering programme

## 1. Introduction

It is the goal of institutes of higher learning (IHL) to put their reputations and their names on the top choices of the best-ranking universities [1]. The task requires a paradigm shift in the entire administration of a higher-learning institution. A good educational framework and systematic plan are needed to make engineering programmes succeed. In Malaysia, teaching and learning in engineering education have evolved drastically with the introduction of output-based education [2–4]. This OBE system focuses more on outcomes and the quality of graduates upon completion of their studies. The institution's ability to generate graduates with the particular domain stated in the programme

educational outcomes is to be expected from this system [5]. PEOs also indicate strategic development in the intended level of career- and professional accomplishment and in the level of achievement of engineering graduates according to the ABET 2011 standard [6]. According to the requirements, PEOs shall be published with the process, results shall be assessed, and evidence from stakeholders' involvement shall be made clear [7]. Malaysian graduates are also expected to capture eight domains set in the qualification framework as their learning outcomes, namely: (1) knowledge; (2) practical skills; (3) social skills and responsibilities; (4) values, attitudes, and professionalism; (5) communication, leadership, and team skills; (6) problem-solving and scientific skills; (7) information management and lifelong-learning skills; and (8) managerial and entrepreneurial skills [8,9]. The IHL must ensure their graduates are equipped with appropriate knowledge and skills to meet clients' needs, and training for sustainability is now recognised as essential. The aim of the study was to identify the most recommended attributes for PEO statements to ensure the programme offers relevance to the current trend, to ensure it was sustainable for any changes in industrial demand, and to ensure it met the stakeholders' requirements.

## 2. The Aim of the Study

The main objective of this study was to determine which graduate attributes must be developed for securing employability in a currently demanded market. The results can be used to evaluate the academic management system by placing the foremost demanded attributes for engineering graduates. This is to ensure the sufficiency for graduates to have adequacy in its breadth and depth as required. Initially, the process of identifying them was performed through a PEO comparative study between universities offering similar EE-engineering programmes. The analysis and the PEO evaluation are important to ensure the right attributes are placed for evaluating the overall academic-management system. This was then followed by commendable action with continuous quality improvement within the academic-management structure. With a good academic-management model in place, the quality of graduates is expected to improve together with additional values. These outcomes are not just to exceed the expectation of learning courses but also to pre-pare graduates to face the actual working environment [10]. The outcomes will transform whatever knowledge was learned into fulfilling the needs of stakeholders, including em-ployers and industries. Moreover, engineering graduates are expected to attain specific domains relevant to their professional field. Therefore, each institution must establish dynamic PEOs aligned with its mission and vision. PEO-statement inputs are derived from internal or external sources such as alumni, students, lecturers, industrial advisory panels (IAPs), employers, and other stakeholders. As a common practice, PEOs are reviewed through analysis and discussion on the inputs with industrial advisory panel members. Another mechanism of obtaining stakeholders' input is through employer questionnaires, industrial training employer surveys, and alumni surveys [11]. An alumni survey, which is used a few years after graduates have completed their studies, is one good example of indirect measurement for assessing PEO achievement [12]. Their career performance is used to benchmark the real performance and effectiveness of the PEO survey [13]. Indirect measurement of PEO achievement is then analysed accordingly to programme learning outcomes (PLOs) attainment for engineering graduates. A PEO–PLO matrix can determine the coverage of the domains required for engineering graduates.

Questionnaires, interviews, discussions, and surveys were used for evaluating PEOs and their current engineering demands. Upon completing a circle of assessment, PEOs will then undergo a preview stage to see the relevancy of their performance indicators, which were set earlier to meet the current needs [14]. In terms of selecting suitable PEOs, several approaches were developed according to expected outcomes from graduates within five years after graduation by professional expertise, innovators, and leadership. PEO outcome results from the mapping exercise were used to formulate an anonymous online questionnaire survey as a measure of the PEOs' attainment [15]. Some institutions even stipulate the PEOs of the programme without empowering the faculty to develop, own,

and adopt a consistent set of measurable PEOs [16]. To some extent, the graphical method of representing PEOs is visualisation, which can complement the standard descriptive item. The graphics enable the representation of target indicators for each component in every PEO statement [17]. Though different approaches listed in a presentation can help, the most important issue is to know how the PEOs were created from their engineering programme structure. By streamlining potential PEOs, it will be a benchmark towards accepting commonly shared attainable graduate attributes.

Comparative PEO studies for engineering institutions are used to analyse the pattern of attributes shared and embedded in improving teaching and learning processes. Similarly, the comparison performed with target-attainment levels for each performance measure provided satisfactory results of the programme [18]. The most-popular PEO attributes among these institutions can become a preference PEO among institutes offering similar engineering programmes. Perhaps the most-significant aspect of PEO comparative studies is to highlight the current trend on the type of skills and the type of engineering personality the industry is looking for. With the outcomes, each university can review and start to improve the stakeholders' demand with continuous-quality-improvement (CQI) activities [9,19]. Studies on engineering, including the results on engineering graduates' skills [20], have become eminent and have expanded to other Middle East and African regions. In Malaysia, besides meeting EAC and MQF requirements, lately, conceive–design–implement–operate (CDIO)-initiative approaches have also received positive feedback in evaluating engineering programmes [21]. Various inputs, opinions, and views from different stakeholders tend to improve the programmes and produce better PEO choices. Moreover, engineering institutions can explore other options and can share their engineering education best practices. This is to ensure that the programmes offered are relevant and sustainable to meet the current demand.

### 3. Methodology

The methodology used in this PEO comparative study was based on electrical and electronic engineering programme-educational-objectives (PEO) statement data, which were taken from the official website of each IHL. A total of 30 Malaysian institutions of higher learning offering EE-engineering programmes were selected. These recognised engineering institutions were registered earlier under the Ministry of Higher Education (MoHE), which is currently known as the Ministry of Education (MoE). The PEO definition stated in the Engineering Accreditation Council (EAC) manual was used throughout this comparative study involving accredited engineering universities. The term used is compatible with that of ABET and other recognised engineering accreditation bodies.

Malaysian higher education has four categories: public universities, private universities, foreign branches, and college universities. Firstly, public universities are divided into research universities (RU) or non-research universities (NRU). Secondly, the private universities (PU) are divided into two groups: semi-government universities (SGU) or full private universities (FPU). Thirdly, institutions that have their subsidiaries in Malaysia are known as foreign-branch universities (FBU). Lastly, there is a category of institutes offering engineering courses known as university colleges (UC), and this category was selected. To compare them globally, another twenty (20) reputable universities offering Bachelor of Electrical and Electronic Engineering programmes were added from the USA, UK, Australia/New Zealand, and Asian regions, thus making up a total of fifty (50) universities accounted for in this global PEO comparative study (refer to Table 1).

**Table 1.** Malaysian universities and university colleges together with other global regions' reputable universities offering Bachelor of Electrical and Electronic Engineering programmes.

| Public (RU) | Public (NRU) | Private (SGU) | Private (FPU) | Foreign Branch Univ (FBU) | Univ Colleges (UC) | US | UK | Aust/NZ | Asian |
|---|---|---|---|---|---|---|---|---|---|
| UM | UTeM | UNITEN | Taylor's Univ | Curtin | TARUC | MIT | Cambr | Melbourne Unv | Natl Univ Singapore |
| UKM | UTHM | UTP | AIMST Univ | Swinburne | KLMUC | Stamford | Oxford | Australia Natl Univ | UST HKong |
| USM | UPNM | MMU | Inti Univ | Monash | KDUC | UC Berkeley | Imp College | UNSW | Tsing Univ China |
| UPM | UniMAP | Unisel | SEGi Univ | Nottingham | Stamford UC | Caltech Unv | Manchester Univ | Sydney Univ | Tokyo U Japan |
| UTM | UMP | UniKL | UCSI | | | Univ Of Illinois | College London | Auckland Uni, NZ | Seoul Natl Univ-Korea |
| | | UiTM | UTAR | | | | | | |
| **5** | 5 | 6 | 6 | 4 | 4 | 5 | 5 | 5 | 5 |
| Total Numbers of Universities = 50 | | | | | | | | | |

As for PEO statements, a list of twenty-eight (28) attribute keywords was identified from PEO statements gathered. These attributes are itemised and coded using PEO01 until PEO28 (Refer to Table 2). These PEO key attributes were taken directly from each university's Electrical and Electronics Engineering Department official websites. The domain attributes focus more on the above courses offered in these institutions of higher learning.

**Table 2.** Listing of PEO01-PEO10 (matches EAC Manual) while PEO11 and above are additional.

| Code | Key Attributes Term | Code | Key Attributes Term |
|---|---|---|---|
| PEO01 | Knowledgeable in engineering | PEO15 | Career-building/personal qualities development |
| PEO02 | Communications/interpersonal skills | PEO16 | Creative and innovation engineers |
| PEO03 | Competency in field/tech expert/Registered | PEO17 | Continue Education/Professional development training |
| PEO04 | Problem-solving | PEO18 | Carrying research and development work |
| PEO05 | Know-how skills/productive/system approach | PEO19 | Practice/contributes expertise (social contribution) |
| PEO06 | Sustainability development awareness | PEO20 | Possess military leadership/profession |
| PEO07 | High ethics values and Professionalism | PEO21 | Possess engineering management |
| PEO08 | Individual Leadership/Teamwork | PEO22 | Proficiency of soft skills |
| PEO09 | Cultural, environment, economic, safety, cost/Global Impact/changes | PEO23 | Forefront of technology |
| PEO10 | Lifelong learning | PEO24 | Commercialised products |
| PEO11 | National Inspirational/belonging | PEO25 | Project management/finance |
| PEO12 | Focus on niche/specialised area | PEO26 | Globalisation/international |
| PEO13 | Entrepreneurship | PEO27 | Information technology |
| PEO14 | Multidisciplinary/skills engineers | PEO28 | Others |

PEO mapping for the electrical and electronic engineering department from selected universities are compared with the listed PEO domain. Data are collected and summed up based on PEO itemised code. Each attribute is then plotted in matrix form against attributes term found in their respective Electrical and electronic engineering PEO statement. Histograms are used to represent graphical displays for better visualisation for identifying frequency of attributes which occur in each itemised PEO codes. Meanwhile, the Venn diagram is used to illustrate the attributes' grouping categories either as common, sharing, or uniqueness associated to each attainment attribute. The common group refers to those attributes that are dominantly used by the majority. The sharing group refers to those attributes that are shared between two or more university stakeholders. Finally, the uniqueness group is defined as any extraordinary attributes that are not shared and become their own independent groups in fulfilling their missions and visions. Data obtained from the selected institution are then analysed visually according to the summation of PEO attribute keywords. Analysis of data shows PEO domain attributes had a different niche

and different area of interest. Listed below are 10 categories of selected EE-engineering universities used in comparing the PEO statement attribute keyword.

(a)     PEO comparison among Malaysian public research universities (RU)
(b)     PEO comparison among Malaysian public non-research universities (NRU)
(c)     PEO comparison among Malaysian private universities (either SGU or FPU)
(d)     PEO comparison among Malaysian university colleges (UC)
(e)     PEO comparison among Malaysian foreign branch university (FBU)
(f)     PEO comparison among reputable United States universities (US)
(g)     PEO comparison among reputable United Kingdom universities (UK)
(h)     PEO comparison between reputable Australia and New Zealand universities (Australia/NZ)
(i)     PEO comparison among selected reputable Asian universities (Singapore, Japan, Korea, HK, China)
(j)     PEO overall analysis for electrical and electronics (EE) engineering programmes

### 3.1. Malaysian Public Research Universities (RU)

Public universities with research status in Malaysia have to be active in generating new ideas, especially in the latest research niche such as renewable energy, optics, medicine, etc. As seen in Table 3, the most common PEO statement used by the public's research university is to incorporate creativity and innovation (PEO16) in engineering careers. Apart from aiming the niche for the institution, public research university activities are also focusing on the high ethics and professionalism (PEO07) of their research work.

**Table 3.** Malaysian research university attributes.

| Classification | POs | Attributes of Research University (RU) |
|---|---|---|
| Cluster (Dominant) | PEO7 | High ethical values and Professionalism |
| | PEO16 | Creative and innovative engineers |
| Sharing (Stakeholder Input) | PEO1 | Knowledgeable in engineering |
| | PEO3 | Competency in field/tech expert/registered |
| | PEO6 | Sustainability development awareness |
| | PEO8 | Individual Leadership/teamwork |
| | PEO18 | Carrying research and development work |
| Uniqueness (M&V) | PEO5 | Know-how skills/productive/system approach |
| | PEO9 | Promote general sustainability in cultural, environmental, economic, safety, cost/global Impact/changes |
| | PEO10 | Lifelong learning |
| | PEO11 | Inspirational of Malaysia |
| | PEO15 | Career-building/personal quality development |
| | PEO17 | Continue education/professional development training |
| | PEO22 | Proficiency of soft skills |
| | PEO23 | Forefront of technology |

Research universities must ensure all research outcomes, ideas, patents, plagiarism, and copyright issues are in order. On top of these, research universities are inspired to produce more engineers who have leadership capabilities, sustainability awareness, and competency in their respective engineering fields. In preserving a reputation as a research university, the institution is expected to produce quality with innovative outcomes from these graduates. Many awards and academic achievements are obtained through good results of research and development work. With such engineering education transformation, research universities have directly contributed to ensuring success by appearing in the international arena with the latest innovations and distinguishable research outcomes. From the data (Figure 1), the result shows research universities shun away from a wider scope of engineering areas but concentrated more on producing higher-value research papers and outcomes. Details of the PEO selections of Malaysian research university have

been published in one of the writers' submitted journals [22]. The histogram chart for research universities (RU) offering an EE programme (UKM, UM, USM, UPM, UTM) is depicted in Figure 2. Ethics, creativity, and innovation are the most popular attributes among those selected in the list.

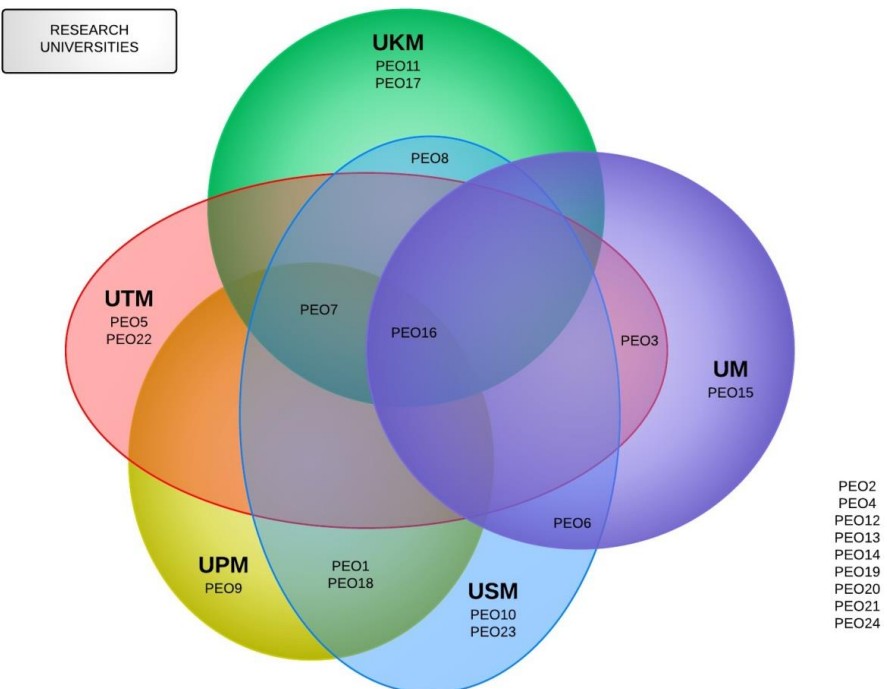

**Figure 1.** PEO for Malaysian research universities (RU).

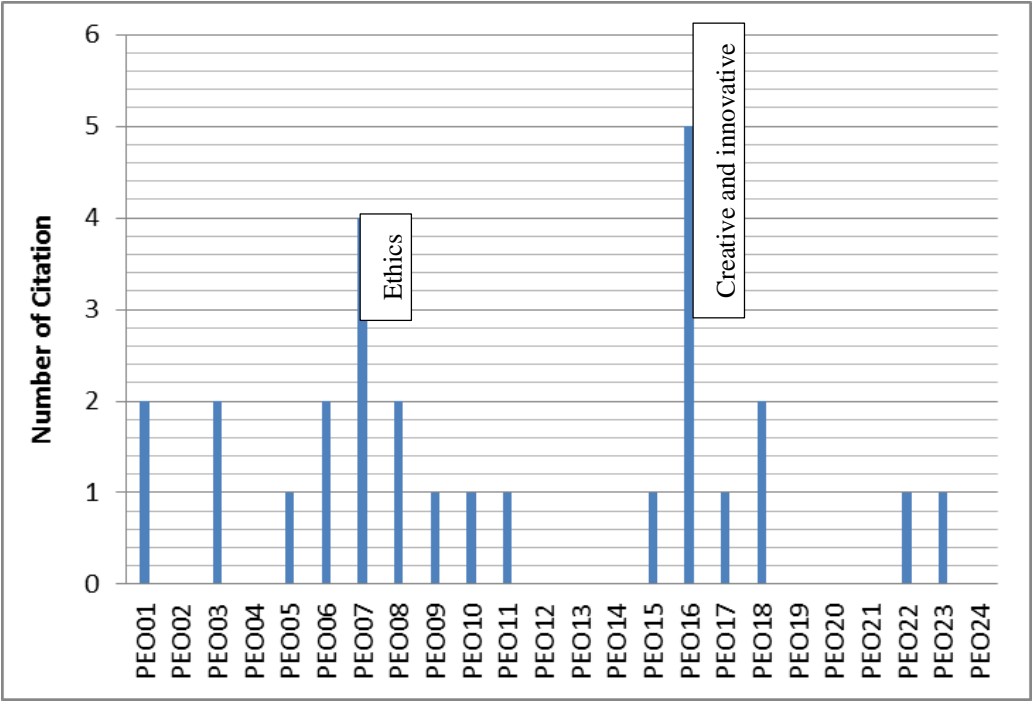

**Figure 2.** Histogram chart for research universities (RU) offering EE programme (UKM, UM, USM, UPM, UTM).

### 3.2. Malaysian Public Universities (Non-Research Universities)

There is a plain distinction between public research universities and non-research public universities, which are more focused. Differences in PEO attributes are made clear when focusing on the university's objectives (refer to Table 4). Most of the non-research universities focus on a specific goal aligned with the mission and vision of the university. Competency in the field of technical expertise is the highest for each non-research university. The technical competency covers an about 80% ratio including those of UTeM, UPNM, UMP, and UniMAP. Each university listed has its own special technical competencies (PEO3) specialising in technology engineering fields. This involved manufacturing, agriculture, construction, electronics, and service sectors. Apart from producing competent graduates, non-research universities also emphasise leadership and teamwork (PEO8) as another focal attribute.

**Table 4.** Malaysia non-research university attributes.

| Classification | POs | Attributes of Non-Research Universities |
|---|---|---|
| Cluster (Dominant) | PEO3 | Competency in field/tech expert/registered |
| | PEO8 | Individual leadership/teamwork |
| Sharing (Stakeholder Input) | PEO7 | High ethical values and professionalism |
| | PEO10 | Lifelong learning |
| | PEO18 | Carrying research and development work |
| Uniqueness (M&V) | PEO1 | Knowledgeable in engineering |
| | PEO13 | Entrepreneurship |
| | PEO15 | Career-building/personal qualities development |
| | PEO16 | Creative and innovative engineers |
| | PEO17 | Continue education/professional development training |
| | PEO19 | Practice/contributes expertise |
| | PEO20 | Possess military leadership/profession |

Engineering courses offered involved both theory and practical 'hands on' aspects as part of the learning programme to consolidated balance attainment among graduates an electrical and electronic engineering programme. In general, non-research universities run their programmes by focusing more on actual industrial needs and utilising their own internal engineering technologies niche, as illustrated in Figure 3. In carrying out technical work, students also upheld high ethical and professional standards and extending lifelong learning in engineering technologies. Non-research universities not only have engineering theory but lean more towards the hands-on experiences to focus on achieving competency in their field of expertise. A PEO histogram chart for non-research universities (NRU) offering EE programme (UTeM, UTHM, UniMAP, UMP, UPNM) is depicted in Figure 4.

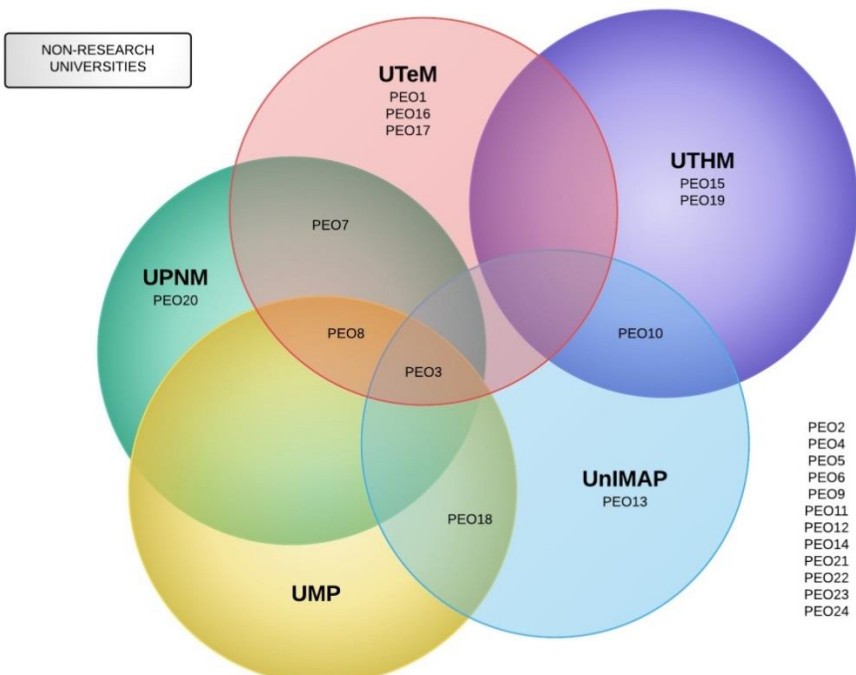

**Figure 3.** PEO for Malaysian non-research universities (NRU).

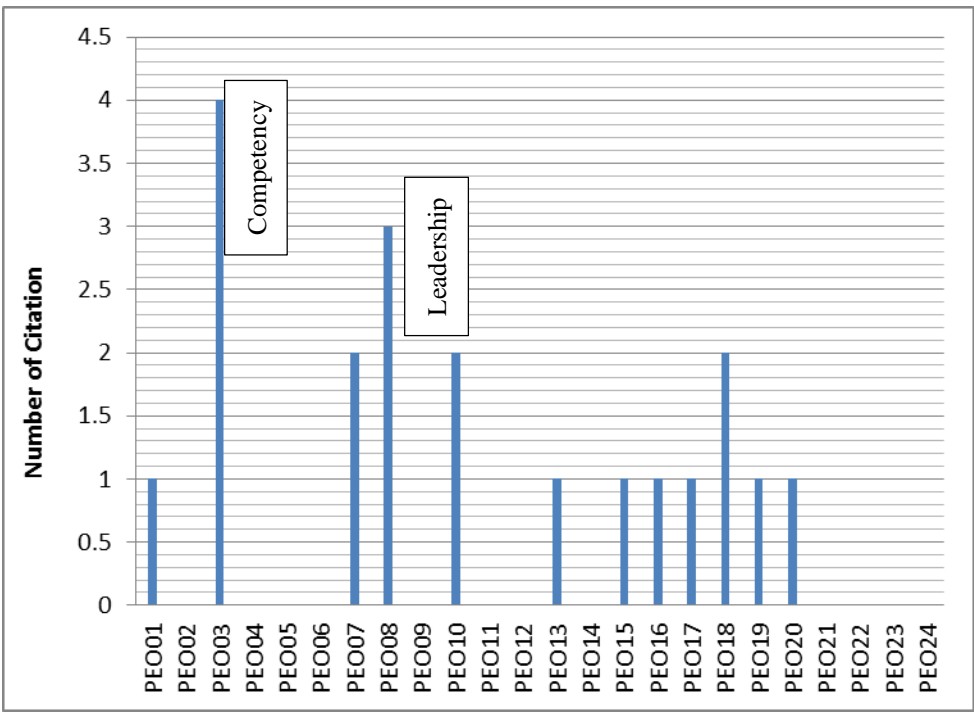

**Figure 4.** PEO histogram chart for non-research universities (NRU) offering EE programme (UteM, UTHM, UniMAP, UMP, UPNM).

### 3.3. Malaysian Private Universities (PU)

From Table 5, the PEO that dominates the most in private institution for electrical and electronic engineering is competency (PEO03) and the ability to develop leadership and teamwork (PEO08) in an organisation by promoting teamwork in a more challenging position. For private semi-government universities (SGU), the emphasis is more on competency, teamwork leadership, and on multidisciplinary skills. Fully private institutions (FPU), however, have chosen leadership and teamwork as their central pillar in PEO statement

choices. Both of these domains, as highlighted by the universities, are directly linked to the leadership qualities of the engineering graduates.

**Table 5.** Malaysian private university attributes.

| Classification | Pos | Attributes of Private Universities (SGU/FPU) |
|---|---|---|
| Cluster (Dominant) | PEO3 | Competency in field/tech expert/registered |
| | PEO8 | Individual leadership/teamwork |
| Sharing (Stakeholder Input) | PEO2 | Communications/interpersonal skills |
| | PEO7 | High ethics values and professionalism |
| | PEO9 | Promote general sustainability in cultural, environmental, economic, safety, cost/global Impact/changes |
| | PEO10 | Lifelong learning |
| | PEO14 | Multidisciplinary/skills engineers |
| | PEO17 | Continue education/professional development training |
| | PEO18 | Carrying research and development work |
| | PEO19 | Practice/contributes expertise |
| Uniqueness (M&V) | PEO1 | Knowledgeable in engineering |
| | PEO4 | Problem-solving |
| | PEO6 | Sustainability development awareness |
| | PEO12 | Focus on niche/specialised area |
| | PEO13 | Entrepreneurship |
| | PEO16 | Creative and innovation engineers |
| | PEO22 | Proficiency soft skills |

Private universities are capable of producing a fresh generation of graduates not only technically competent but also emphasising on the quality aspects of the employability of their graduates. In order to be relevant to the job market, few additional attributes could supplement some of the gap left by public universities in retaining the employability among engineering graduates. Overall, most private universities in Malaysia do not focus too much on research or anything to do with highly intensifying research and development projects in IHL, instead focusing more on how to market their graduates with correct skills such as leadership and competency in their fields of study (see Figures 5 and 6). A PEO histogram chart for private universities (SGU/FPU) offering an EE engineering programme (UNITEN, UTP, MMU, UNISEL, UNIKL, UITM, TAYLOR, AIMST, INTI, SEGI, UCSI, UTAR) is depicted in Figure 7.

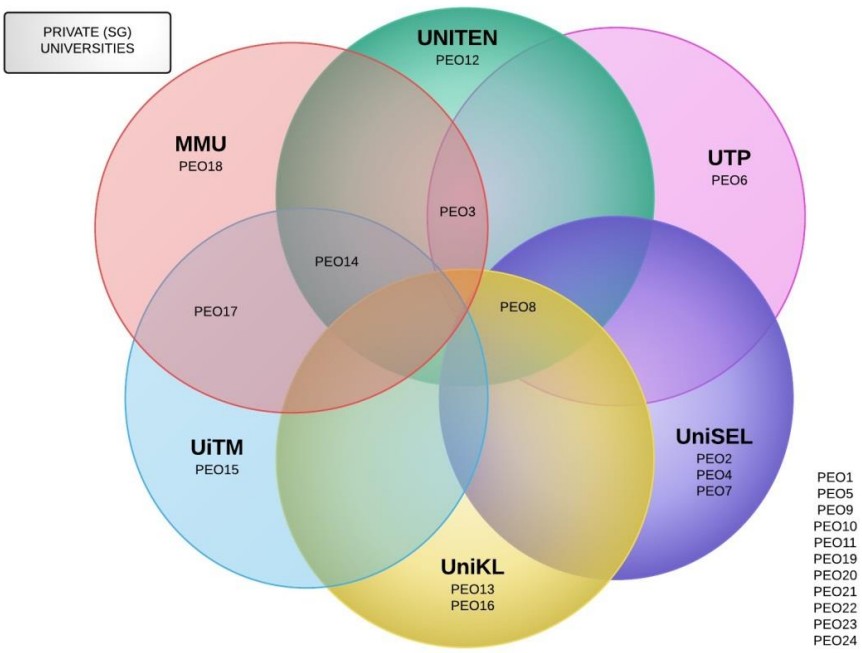

**Figure 5.** PEO for Malaysian private semi-government universities (SGU).

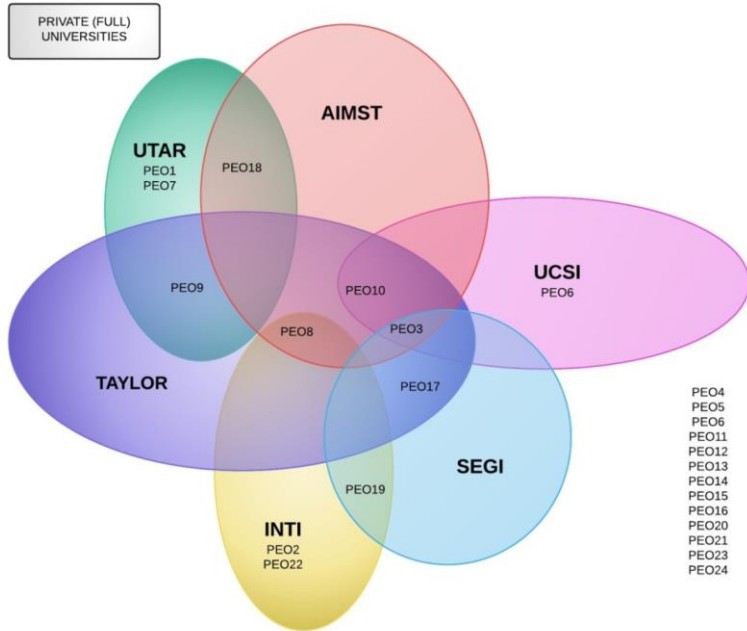

**Figure 6.** PEO for Malaysian fully private universities (FPU).

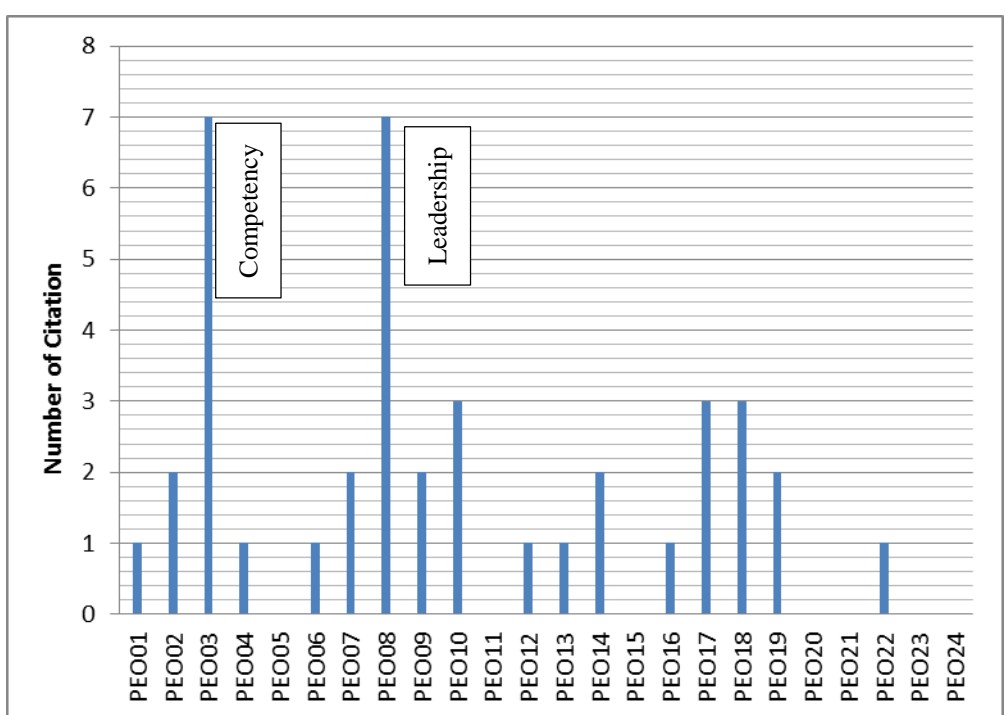

**Figure 7.** PEO histogram chart for private universities (SGU/FPU) offering an EE engineering programme (UNITEN, UTP, MMU, UNISEL, UNIKL, UITM, TAYLOR, AIMST, INTI, SEGI, UCSI, UTAR).

*3.4. Malaysian University Colleges (UC)*

There is no one dominant choice of PEO statements chosen by most institutions of higher learning under university colleges categories in Malaysia (refer to Table 6). Engineering knowledge (PEO1) and competency (PEO3) are more of a complacency for their attributes' choices. As a matter fact, each university colleges have their own way of defining PEO based on each university college's mission and vision statements. They use their own strategy of producing engineering graduates in compliance with the conditions set by the Engineering Accreditation Council (EAC) to produce quality engineering graduates. The comparison study found that University College's institution is focused more on domains that are not conventionally used. For example, they emphasise multidisciplinary skills (PEO14). Additionally, some colleges added the PEO statement that leads to expanding engineers' capability in the field of entrepreneurship (PEO13) and communication skills (PEO02).

The strategy being used is in helping their graduates to be accepted and in venturing into new engineering opportunities in a market that is already saturated and competitive. This phenomenon brings into the job market a larger group of engineers with supplementary skills aside from electrical and electronic know-how. Obviously, each university college has its diverse programme targets for their graduates. The PEO statement needs to reflect shareholders' needs to compete healthily with other public and private institutions. With action taken, college universities are hoping to benefit their graduates to be more progressive and to quickly adapt to market changes. Overall, similar to that of private institutions, the university college is more distanced from research-based activities but instead focusing on the adaptability of engineering graduates with multidisciplinary skills (entrepreneur, communication, specialised management, commercialised product, etc.) (see Figure 8). A PEO histogram chart for university colleges PEO offering an EE engineering programme (KDU, STAMFORD, TARUC, KLMUC), as depicted in Figure 9.

**Table 6.** Malaysian university college attributes.

| Classification | Pos | Attributes of University Colleges (Ucs) |
| --- | --- | --- |
| **Cluster (Dominant)** | | |
| **Sharing(Stakeholder Input)** | PEO1 | Knowledgeable in engineering |
| | PEO3 | Competency in field/tech expert/registered |
| **Uniqueness (M&V)** | PEO2 | Communications/interpersonal skills |
| | PEO4 | Problem-solving |
| | PEO5 | Know-how skills/productive/system approach |
| | PEO6 | Sustainability development awareness |
| | PEO7 | High ethics values and professionalism |
| | PEO8 | Individual leadership/teamwork |
| | PEO9 | Promote general sustainability in cultural, environmental, economic, safety, cost/global Impact/changes |
| | PEO12 | Focus on niche/specialised area |
| | PEO13 | Entrepreneurship |
| | PEO14 | Multidisciplinary/skills engineers |
| | PEO16 | Creative and innovation engineers |
| | PEO17 | Continue education/professional development training |
| | PEO21 | Possess engineering management |
| | PEO24 | Commercialised products |

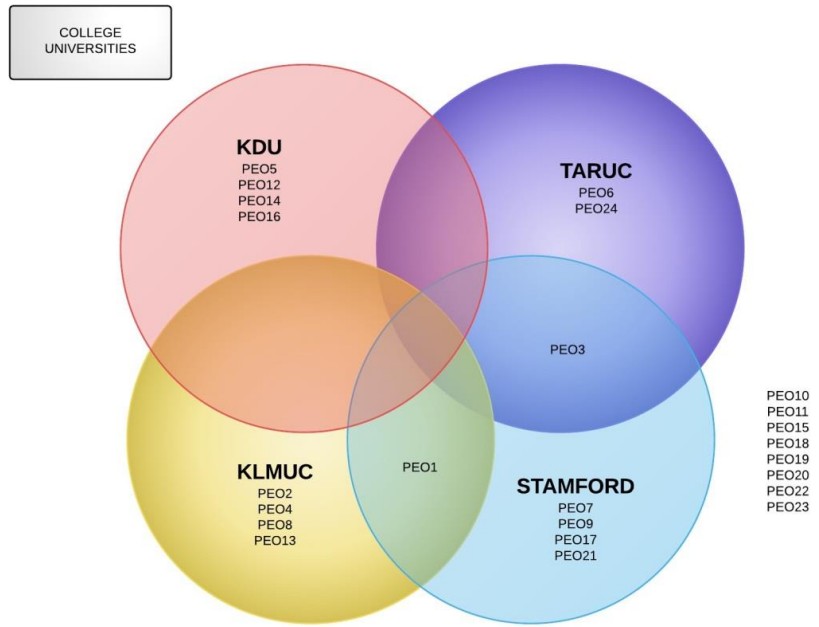

**Figure 8.** PEO for Malaysian university colleges (UC).

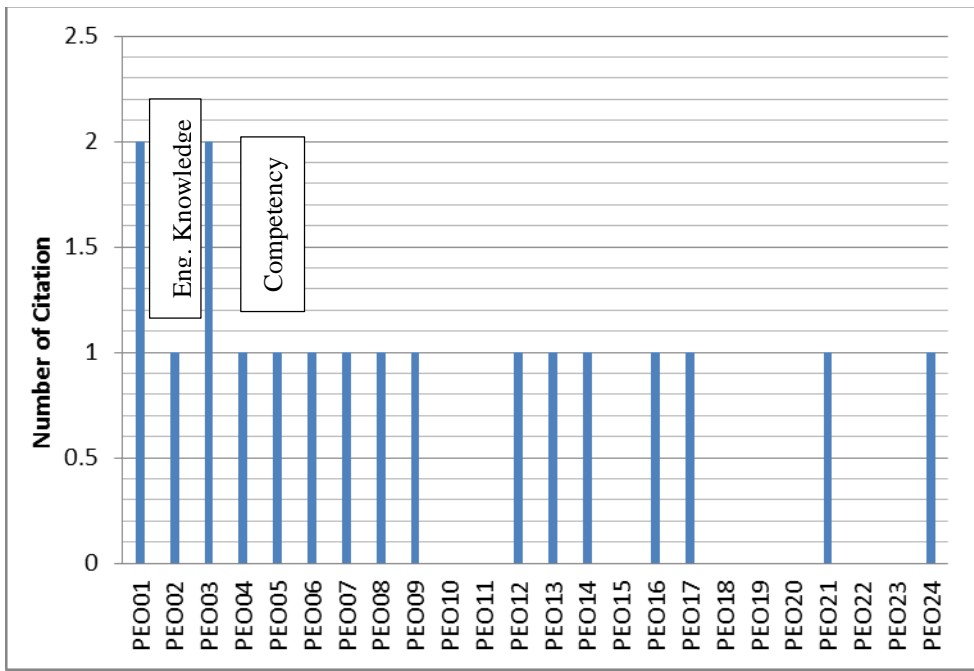

**Figure 9.** PEO histogram chart for university colleges PEO offering an EE engineering programme. (KDU, STAMFORD, TARUC, KLMUC).

*3.5. Malaysian Foreign Branch University (FBU)*

Two PEO attributes are clearly identified from foreign universities with branches in Malaysia. The first area is on developing leadership and teamwork (PEO08) among talented graduates, embedded with the latest engineering knowledge and promoting the engineering environment, economic, safety, and cost impact (PEO09) knowledge needed to assist them in sustaining social development (refer to Table 7).

**Table 7.** Malaysian foreign branch universities attributes.

| Classification | Pos | Attributes of Foreign Branch Universities (FBU) |
| --- | --- | --- |
| Cluster (Dominant) | PEO8 | Individual leadership/teamwork |
| | PEO9 | Promote general sustainability in cultural, environmental, economic, safety, cost/global Impact/changes |
| Sharing(Stakeholder Input) | PEO1 | Knowledgeable in engineering |
| | PEO7 | High ethics values and professionalism |
| Uniqueness (M&V) | PEO3 | Competency in field/tech expert/registered |
| | PEO5 | Know-how skills/productive/system approach |
| | PEO6 | Sustainability development awareness |
| | PEO10 | Lifelong learning |
| | PEO14 | Multidisciplinary/skills engineers |
| | PEO15 | Career-building/personal qualities development |

Investing a large sum of money is expected to generate quality education for foreign universities to have branches in Malaysia. With experienced educators and expatriates brought into their Malaysian campuses, it strengthened the university in providing quality courses. With this collaboration, the plan could help to uplift the graduates' quality globally in order to maintain the reputation of their primary institutions. It is not surprising that the quality of education is designed to produce graduates who are not only knowledgeable but also the future leaders of their respective organisations. Many foreign universities offering engineering programmes in Malaysia give the opportunity for Malaysians to gain experience at their main campuses overseas. Almost all foreign branches had differentiated

themselves and their prestigious graduates. Overall, PEO for these universities seems to have similar main pillars with research and non-research universities such as leadership and teamwork, and ethics. However, a particular emphasis on sustainability on the global environment is notable (see Figure 10). Being established in the higher education field worldwide justified their establishment to focus more on global issues. A histogram chart for the foreign universities with branches in Malaysia offering an EE programme is depicted in Figure 11.

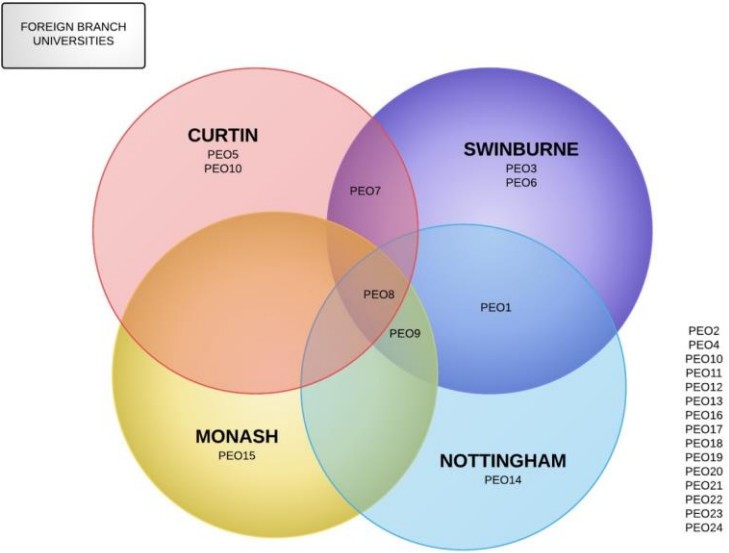

**Figure 10.** PEO for foreign universities with branches in Malaysia.

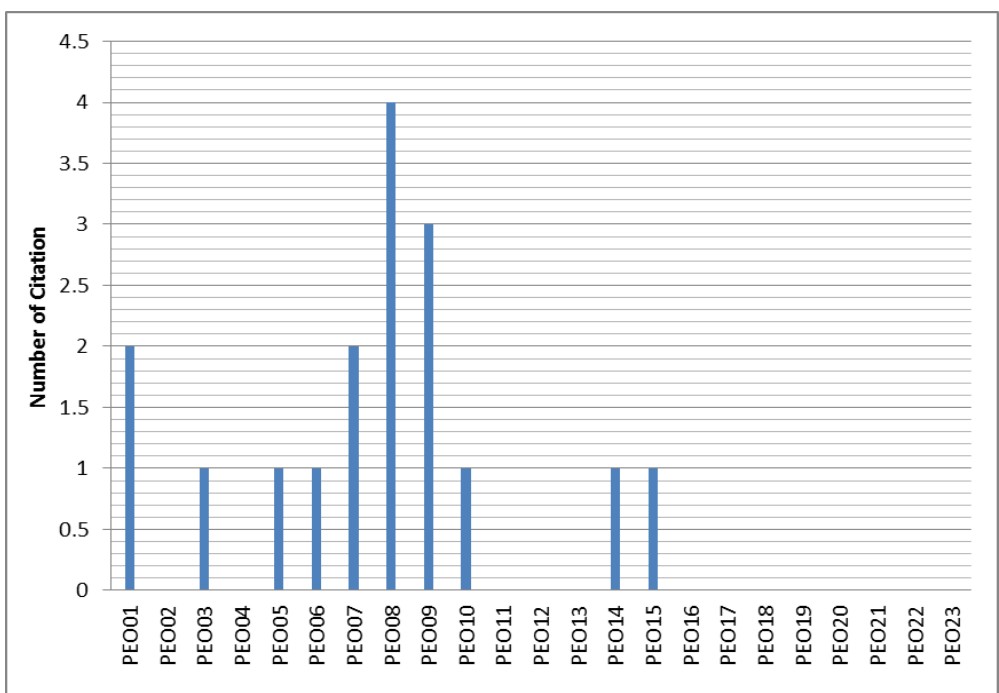

**Figure 11.** PEO histogram chart for the foreign branch universities offering EE programme (Curtin, Swinburne, Monash, Nottingham).

*3.6. Global Universities Views on EE Engineering Graduates*

Looking at global regional views, electrical and electronic engineering graduates prefer to have some essential skills required for their profession. The following summary

represents some differences in perception gathered from what other world reputable engineering universities have when it comes to preparing holistic attributes for future electrical and electronic (EE) engineers.

### 3.6.1. United States of America (USA)

Five reputable well-known universities offering engineering programmes are selected for PEO comparative studies based on their expected objectives. The selected universities are the Massachusetts Institute of Technology (MIT), Stamford University, University of California at Berkeley, California Institute of Technology (CalTech), and the University of Illinois (refer to Figure 12). These reputable engineering universities were chosen following their high personal contribution in the marketability and employability of EE engineering graduates. Leadership among engineers (PEO8) was given a top priority among selected programmes objectives. The other most preferred PEO attributes for engineering are on continuous education for graduates' competencies/skills (PEO3), and not least on engineering professionalism and ethical values in society (PEO7). Most of the US engineering universities applied ABET programme outcomes as their preferred standard. PEO choice might defer according to each institution specialty and the focus in a particular niche.

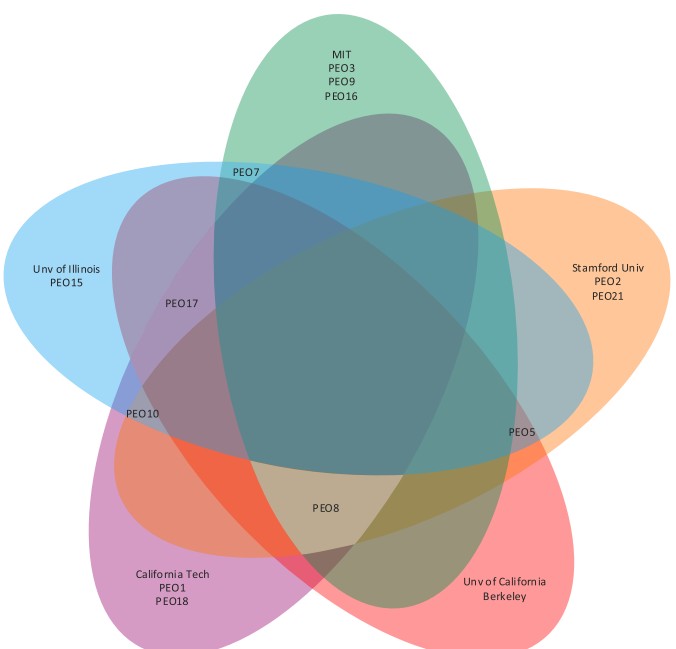

**Figure 12.** PEO used by the United States of America (USA)'s reputable engineering universities.

### 3.6.2. United Kingdom (UK)

Similarly, five reputable and well-known UK universities offering engineering programmes are selected for PEO comparative studies. The universities selected are Cambridge, Oxford, Imperial College of London, the University of Manchester, and the University of College London (refer to Figure 13). All these five prestigious UK institutions are among the best student placements for engineering programmes in the UK. Among the highest PEO attributes emphasis for EE engineering graduates are leadership and teamwork (PEO8) as well as on student competency (PEO3). Additionally, many UK engineering universities are focused more on competitiveness and continuing education/professional development training (PEO17) to excel in their engineering careers. The other PEO attributes emphasised are personal development qualities (PEO15) in order to prepare them in facing challenges within their engineering careers.

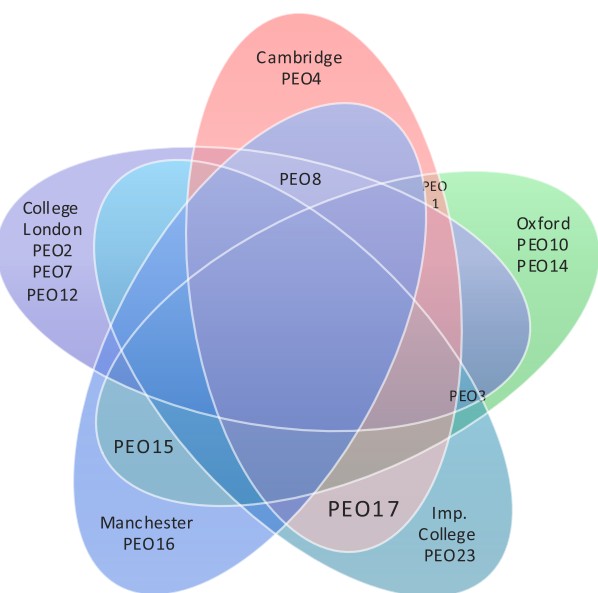

**Figure 13.** PEO used by the United Kingdom's reputable engineering universities.

### 3.6.3. Australia and New Zealand

The five reputable engineering universities of Australia and New Zealand that were chosen are the University of Melbourne, Australia National University, the University of New South Wales, the University of Sydney, and the University of Auckland (refer to Figure 14). All these institutions are considered to be among the best universities in the southern global hemisphere region offering EE engineering programmes. Some of their most remarkable PEO attributes are in engineering knowledge (PEO1). Other high priorities include graduates with great interpersonal and communication skills (PEO2), problem-solving skills (PEO4) and graduates possessing high ethics and professionalism (PEO7). Meanwhile, the application of the CDIO concept in the engineering education framework was adopted among Australian engineering institutions. Those listed PEO attributes are given top preferences in almost all Australian and closely associated institutions offering engineering programmes.

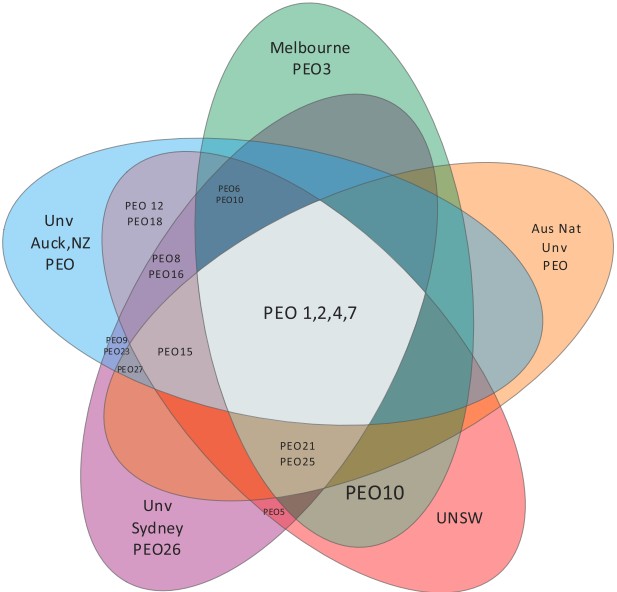

**Figure 14.** PEO used by Australia and New Zealand's reputable engineering universities.

### 3.6.4. Other Asian Universities

Five best-known institutions of higher learning in the Asian region offering engineering programmes are chosen for this PEO comparative study, which are the National University of Singapore, the University of Science and Technology, Hong Kong, Tsinghua University, China, University of Tokyo, Japan and Seoul National University, Korea (refer to Figure 15). All these prestigious universities in Asia are among the best in the region offering EE engineering programmes in their campuses. From the comparative study outcomes, it is clear that the most sought-after traits of engineers produced based on PEO statements are leadership and teamwork (PEO8). On top of the two attributes, work ethics/professionalism (PEO7) and continuous lifelong education (PEO10) were also given priority. Most of the Asian engineering graduates are capable of taking the challenge in facing current emerging issues and its impact on society. Most Asian engineering universities are focused on issues related to environment and engineering impact (PEO9). Another focal point is developing technical soft skills (PEO22) in facing globalisation challenges. To remain competitive, these reputable Asian universities are developing engineers as talented scientists and researchers in their respective engineering fields.

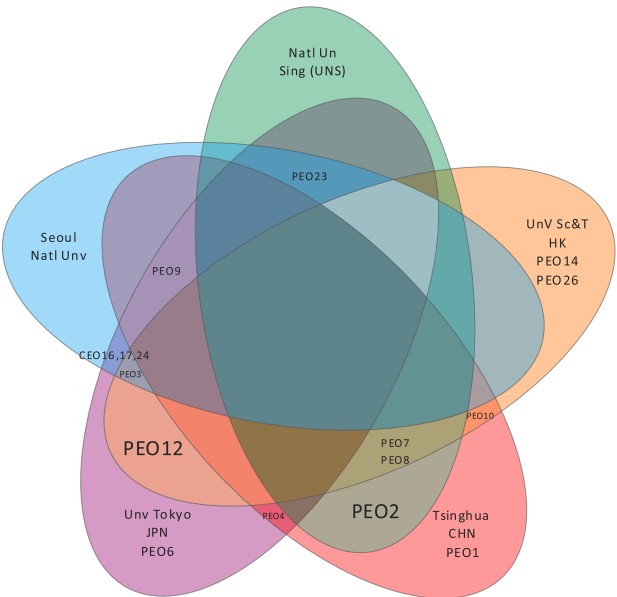

**Figure 15.** PEO statement used by selected reputable Asian engineering universities.

## 4. Results and Analysis

### 4.1. Overall Analysis for Electrical and Electronics (EE) Programme from Global Perspectives

Figure 16 shows PEO attribute keywords chosen among selected global institutions of higher learning offering electrical and electronics engineering programmes. In summary, the overall result shows that the most popular domain chosen by reputable universities globally (USA, UK, Australia/New Zealand, and selected reputable Asian universities) are as follows:

1. Building up leadership capability among EE engineering graduates (PEO08);
2. High ethical values among EE engineering graduates especially on issues related to environments, sustainability, safety, health, and societal issues (PEO07);
3. The importance of obtaining competencies and technical qualifications among EE engineers towards becoming professionals (PEO3); and
4. Emphasis on personal development (PEO15) among EE graduates in those reputable engineering universities.

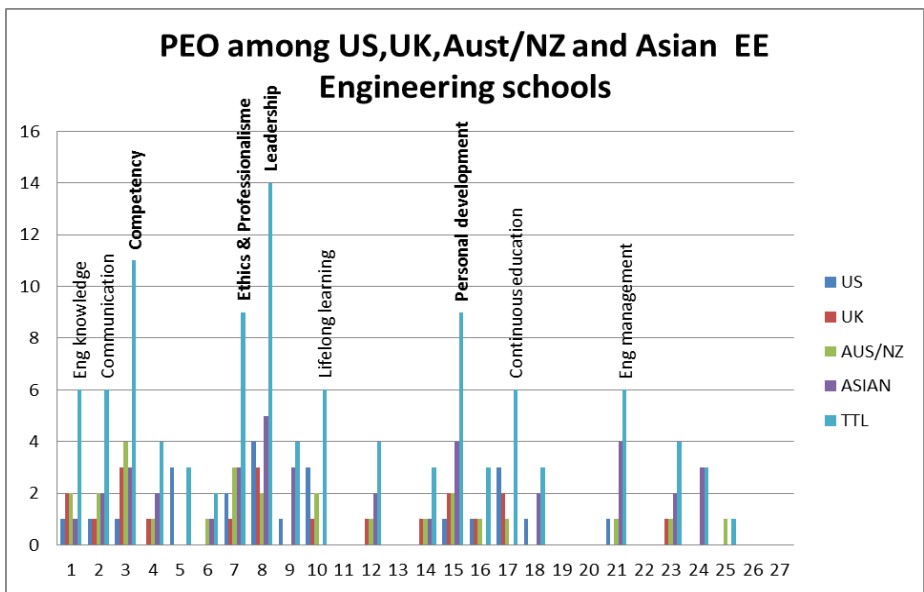

**Figure 16.** Global PEO attribute keywords chosen mostly by universities offering EE engineering programmes in the USA, UK, Australia/New Zealand, and Asian universities.

Table 8 illustrates the preferred PEO attributes choice of which the author has segregated them into four different scale categories.

**Table 8.** List of attribute keywords chosen in EE engineering programmes educational objectives (PEO) statement.

| Classification | PEO | Attribute Keywords | World Region |
|---|---|---|---|
| Most Popular Domain (External Factor) >8 | PEO3 | Competency in field/tech expert/registered | US/UK/AUS/NZ/ASIAN |
| | PEO7 | High ethics values and Professionalism | US/UK/AUS/NZ/ASIAN |
| | PEO8 | Individual leadership/teamwork | US/UK/AUS/NZ/ASIAN |
| | PEO15 | Career-building/personal qualities development | US/UK/AUS/NZ/ASIAN |
| Highly Popular Domain (External Factor and Niches/Cluster) 3 < x < 8 | PEO1 | Knowledgeable in engineering | US/UK/AUS/NZ/ASIAN |
| | PEO2 | Communications/interpersonal skills | US/UK/AUS/NZ/ASIAN |
| | PEO4 | Problem-solving | UK/AUS/NZ/ASIAN |
| | PEO9 | Promote general sustainability in cultural, environmental, economic, safety, cost/global Impact/changes | UK/ASIAN |
| | PEO10 | Lifelong learning | US/UK/AUS/NZ |
| | PEO12 | Focus on niche/specialised area | UK/AUS/NZ/ASIAN |
| | PEO17 | Continue education/professional development training | US/UK/AUS/NZ |
| | PEO21 | Possess engineering management | US/AUS/NZ/ASIAN |
| | PEO23 | Forefront of technology | UK/AUS/NZ/ASIAN |
| Less Popular Domain (Internal Factor/University Choice) 1 < x < 4 | PEO5 | Know-how skills/productive/system approach | US |
| | PEO6 | Sustainability development awareness | AUS/NZ/ASIAN |
| | PEO14 | Multidisciplinary/skills engineers | UK/AUS/NZ/ASIAN |
| | PEO16 | Creative and innovation engineers | US/UK/AUS/NZ |
| | PEO18 | Carrying research and development work | US/ASIAN |
| | PEO24 | Commercialised products | ASIAN |
| Unique Domain (M&V) <2 | PEO11 | Inspirational of Malaysia | |
| | PEO13 | Entrepreneurship | |
| | PEO19 | Practice/contributes expertise | |
| | PEO20 | Possess military leadership/profession | |
| | PEO22 | Proficiency in soft skills | |
| | PEO25 | Project management/finance | AUS/NZ |
| | PEO26 | Globalisation/international | |
| | PEO27 | Information technology | |
| | PEO28 | Others | |

1. Most Popular Domain (External Factor) represents total PEO chosen keyword frequency of 8 and above. This category is considered as the most popular choice among institutions. Normally, it is influenced by external factors based on each institutional choice of PEO statements.

2.  Highly Popular Domain (External Factor and Niches/Cluster) represents total chosen keyword frequency between 4 and 8. This category is considered as highly popular and commonly acceptable PEO attribute keywords among institutions of higher learning. It is influenced by external factors and some representative with its niche and focuses on their cluster area.

3.  Less Popular Domain (Internal Factor/University Choice) represents the total chosen keyword frequency between 1 and 4. This category is considered less popular compared to earlier PEO attribute keywords. However, it has its own values and domains, which may be added to each institution's reputable choices.

4.  Unique Domain (M&V) represents a chosen keyword frequency of less than 2. This category is considered the least preferred choice and is considered a unique attribute for each selected university. It is seldom used to meet the university's vision and mission.

*4.2. Overall PEO Statement for Electrical and Electronics (EE) Engineering Programmes in Malaysia*

Based on the Venn diagram shown in Figure 17, there are four main attributes in the PEO statement identified for electrical and electronic engineering programmes in Malaysia. The first attribute or trait objectives is to achieve qualifications and competency (PEO03) among EE engineering graduates. Securing this attribute will ensure that the continuity in their career as expert engineers be retained and maintained. The second attribute is to achieve and grow their capability of becoming a leader (PEO08) in their professional working groups. The trait of leadership will ensure graduates' credibility and confidence in managing technical matters in various EE engineering organisations. Thirdly, EE engineers are expected to put emphasis on ethics (PEO07) in carrying out their duty as trained engineers. Finally, there is also an emphasis on acquiring engineering knowledge (PEO1) among them. In summary, these selected attributes clearly demonstrate that the EE engineering programme is very much concerned with producing engineers who can lead the organisation with engineers' high moral understanding, professional roles, and responsibility to the society.

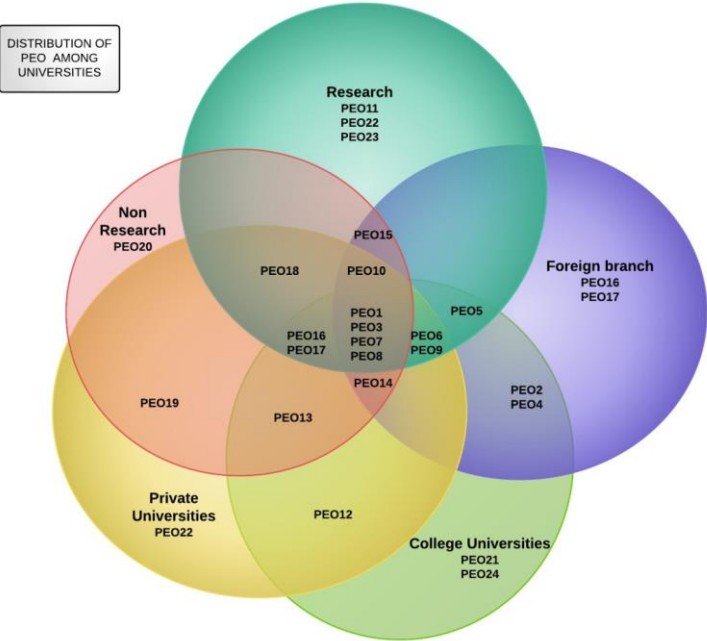

**Figure 17.** PEO statement used in Malaysian universities offering EE engineering programmes (overall).

In Malaysia, the PEO statement was aligned properly as set by the demands of stakeholders. Most universities are demonstrating them by engaging their graduates to achieve

attributes that match with the needs of the industries. From Figure 18 and Table 9, the chosen PEO attributes include knowledge of engineering, competency in the field of expertise or registered, high ethics values and professionalism, and finally, individual leadership/teamwork. Acquiring such domains will improve the quality of EE engineering graduates. Most of these PEO domains come from distinct inputs provided by the stakeholders. EE engineering graduates with a high level of competency are highly in demand to fill up engineering jobs. Good ethics and professionalism are also considered advantageous values serving the engineers working in both public and government sectors. These noble attributes measure the level of responsibility and accountability among local engineers. Malaysian institutions choose individual leadership and teamwork as part of their institutions' pillars. By attaining these attributes, local EE engineers can prepare themselves to become future leaders in their respective organisations.

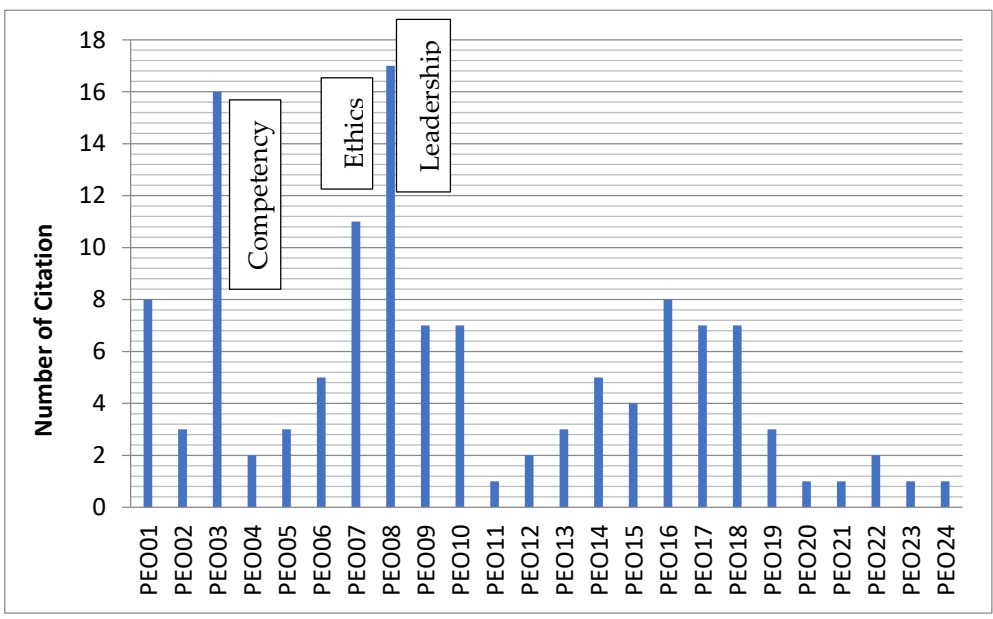

**Figure 18.** Histogram chart among RU, NRU, private, and foreign branch universities offering electrical and electronic (EE) engineering programmes in Malaysia.

**Table 9.** Overall Malaysian EE engineering universities attributes listing.

| Classification | POs | Attributes (Overall Malaysian Universities) | Categories |
|---|---|---|---|
| Most Popular Domain (External Factor) >8 | PEO1 | Knowledgeable in engineering | 12345 |
| | PEO3 | Competency in field/tech expert/registered | 12345 |
| | PEO7 | High ethics values and professionalism | 12345 |
| | PEO8 | Individual leadership/teamwork | 12345 |
| Highly Popular Domain (External Factor and Niches/Cluster) 3 < x < 8 | PEO6 | Sustainability development awareness | 1345 |
| | PEO10 | Lifelong learning | 1235 |
| | PEO14 | Multidisciplinary/skills engineers | 2345 |
| | PEO15 | Career-building/personal qualities development | 125 |
| | PEO16 | Creative and innovation engineers | 1234 |
| | PEO17 | Continue education/professional development training | 1234 |
| | PEO18 | Carrying research and development work | 123 |

**Table 9.** *Cont.*

| Classification | POs | Attributes (Overall Malaysian Universities) | Categories |
|---|---|---|---|
| Less Popular Domain (Internal Factor/University Choice) 1 < x < 4 | PEO2 | Communications/interpersonal skills | 34 |
| | PEO4 | Problem-solving | 34 |
| | PEO5 | Know-how skills/productive/system approach | 145 |
| | PEO12 | Focus on niche/specialised area | 34 |
| | PEO13 | Entrepreneurship | 234 |
| | PEO19 | Practice/contributes expertise | 23 |
| | PEO22 | Proficiency soft skills | 13 |
| Unique Domain (M&V) <2 | PEO11 | Inspirational of Malaysia | 1 |
| | PEO20 | Possess military leadership/profession | 2 |
| | PEO21 | Possess engineering management | 4 |
| | PEO23 | Forefront of technology | 1 |
| | PEO24 | Commercialised products | 4 |

Note: 1 (research university) 2 (non-research university) 3 (private university) 4 (university colleges); 5 (foreign branch universities).

## 5. Discussions and Findings among Malaysian Universities

### 5.1. Research University (RU) Tend to focus on Ethics, Professionalism and Innovation (Cluster)

Research universities are focused mainly on research as part of producing engineers who are capable of carrying out research as well as producing significant publications from their research outcomes. Most research universities strongly emphasise ethical values in their research. Most of the scientific manuscripts from research universities in Malaysia are value-added and can potentially be commercialised by industries as well as for government future planning.

### 5.2. Non-Research Universities (NRU) Tend to Do Research and Innovation Works towards RU Recognition

As for non-research universities in Malaysia, some significant efforts and initiatives are driven to be a part of the larger research community. The attributes are considered as attractive attributes chosen by many non-research universities. Some of the attributes chosen for non-research group clusters will become niches for the institutions. Most attributes are derived from external factors after discussions and interactions with the stakeholders. The outcomes from the discussions with interested parties might be a valuable input for developing these PEO attributes. The potential attributes vary from sound engineering knowledge to sustainability, leadership, lifelong learning, multidisciplinary, career building, creativity, innovation, continuing education, and blended efforts to become part of the culture of communities within the research universities.

### 5.3. Non-Research Universities (NRU) and Private Universities (SGU/FPU) Are Having Similar Trends in Attributes Distribution

There are some similarities identified among non-research universities and private universities in Malaysia. Most of their PEO statements are varied and cover a wide range of attributes. Among the distinguishing items that were discovered is that non-research institutions are also gearing up and preparing themselves to become part of research institutions in the future. There are a number of activities being identified such as carrying R&D, advocating the importance of continuing education and lifelong learning, working towards creation and innovation, as well as engagement in multi-disciplinary and research areas. With the increasing number of research outcomes from non-research and private universities, it indicates that competition between institutions in Malaysia is getting closer. By the year 2020, it is expected that more non-research institutions will join their established research universities to further spearhead the development of engineering education in the country.

### 5.4. Foreign Branch Universities (FBU) Adopt Main Overseas Concepts Such as Sustainability Issues, etc.

As for foreign branch universities established in Malaysia, nothing much can be expected of them since most of the PEO pillars might derive from their main overseas campuses. Among them are attributes used in many advanced countries such as sustainability issues. So much has been mentioned about developing a sustainability concept in developing technology, the environment, and public awareness in general. Apart from producing excellent graduates, many foreign universities in Malaysia are working hard to create a sustainable environment for the benefit of global factors as well as local community development programmes. The idea is in preserving the world as a suitable place to work and study as well as in creating awareness among engineering activities worldwide.

### 5.5. College University (CU) Institutions Use Extensive and Broad Choices of Graduates Attributes

Since the number of statuses of university colleges in Malaysia is still in its infancy, the curriculum structure towards engineering studies is quite general. Each of these institutions might have their way of producing graduates. There is no clear pattern in terms of focusing or selecting niches in their programmes. Programmes cover basic fundamental engineering knowledge, communication, competency, and problem-solving, as well as carrying every single attribute available to match each institutional path. Some university colleges even promote collaboration with overseas universities and colleges by offering twinning programmes. This phenomenon offers their students experience in continuing their studies overseas. However, the students have to prove a sound and upright education track record in order to pursue their studies abroad.

### 5.6. Critical Findings on the Objective for PEO Establishment and Revision

One way for setting good objectives is to carry out gap analysis among PEO of different universities and colleges offering EE engineering. This study can be a benchmark in comparing PEO pillars used in setting up a more realistic and achievable PEOs. The study will recommend the institutions perform a PEO revision and upgrade should there be any significant additions to the attributes to meet the stakeholders' demands. Additionally, not only are the PEOs to be improved but they should also be considered to be at par with the requirements of accreditation bodies as well as to establish a firmer foundation of IHL in enhancing the institutions' mission and vision statements. This revision becomes a more orderly approach in creating the PEO, which is more relevant to the needs of shareholders, as well as improving the graduates' acceptability to meet employability expectations from employers. Most of the current inputs might be derived externally as PEO pillars are becoming more dynamic in nature. With the renewal or rephrasing of PEO among institution of higher learning, a more unimpeachable establishment of PEO statements came out from this comparative study.

## 6. Unique Attributes of PEO among Malaysian Universities

Each institution of higher learning has unique attributes. Besides focusing on leadership skills and ethics, qualifications in the engineering field, UKM had one sole domain area called national aspiration (PEO11). UKM engineers are expected to defend the dignity of graduates in upholding the development of the language, culture, and aspirations of the people of Malaysia on par with graduates from the world's uppermost institutions of higher learning. This spirit of Malaysia also enables the Malay language to be placed as a primary language on par with languages such as Chinese, Korean, Japanese, Indian, Arabic, Spanish, and many others. Although the English language is considered the global language of instruction, upholding the Malay language is one of the unique PEO statements highlighted. Thus, it will boost UKM as a higher education provider in delivering the Malay language globally through engineering education.

Meanwhile, Universiti Malaysia Perlis (UniMAP) took the challenge and task to produce engineers who are competent with entrepreneurship (PEO13) skills. The attributes differ from the multidisciplinary field of engineering by encouraging graduates to venture into the business world. Aside from engineering, other professional engagement skills included are business management, consultation, project management, and R&D activities. The importance of knowledge involved in business within the engineering scope can be further expanded.

Another instance is a project undertaken by Universiti Tenaga Nasional (UNITEN) and Multimedia University (MMU), which are among the few local institutions with significant engineering student intakes in many disciplines. To cater to the job market, programmes intended to produce many engineers with multidisciplinary (PEO24) skills in local and global scenes are being planned. The attributes include equipping them with multi-skills such mastering soft skills, computing, languages, and communication skills. With these additional skills, the graduates will find a wider selection of job opportunities upon graduation. Their engineering capability will undoubtedly increase in line with job competitiveness and higher employability demands among new graduates.

## 7. Significance of PEO Comparative Studies

PEO analysis is one way of seeing how other universities view the statements according to the needs of the institutions. Most standard PEO statements chosen will be at least on par with their competitors or those offering similar programmes in electrical and electronics engineering. A well-established PEO statement indicates an institution's direction, which is in line with much more established organisations worldwide. The essential domains that must be included in the PEO statement for each university are leadership, competency, and principles of ethics.

Competency, ethics in professionalism, and leadership have become the dominant choices of attributes among graduates worldwide, which show the importance of these three attributes in any institution of higher learning. The universities' mission and vision statements depend heavily on the ability to produce the best outcomes out of their institutions. Success in education by delivering knowledge alone does not guarantee the organisations' achievement of excellent graduate results. Other psychomotor and affirmative domains can be added to appear more holistic by providing the best education in any institution of higher learning.

### 7.1. Survivability and Competitiveness of Malaysian Engineering Graduates

In the coming years upon graduation, the graduates need to display performance, skills, and practices in their respective engineering careers. These three attributes are the most essential requirements needed for them to succeed in the future development of their career paths in engineering. Competency, for instance, not only recognises the graduates' capability of performing tasks but also shows the degree of acceptance by industry standard and professional bodies that the engineering work is being carried out for the society by the right person. The level of competency brings about maturity and confidence within the community; the candidates fulfil the requirement and standard of becoming competent within the organisations that they are involved in. With the level of competency instilled among these graduates, the guaranteed profession is intact and remains safe and sound to the society as a whole. Competency attributes demand engineers to be masters in their fields and disciplines. Professional engineers hold a bigger responsibility not only for themselves but also for the sake of the community, environment, and its impact on society.

Similarly, ethical engineering principles play substantial roles in ensuring engineering graduates produced by the institution are made to be responsible, accountable, and abiding by expert guidelines. Graduates should have a good profile and be respected in solving issues related to the planning and sustainability of societal needs. By applying vast and extensive knowledge of design, they are expected to offer their expertise and high ethics towards contributing the best solution for the well-being of the society. Highly regarding

an entity needs to develop within oneself with deep concern towards self-development as professional engineers. This responsibility also links to the ethical issues in engineering in order to preserve a high level of professionalism among practicing engineers. Their survival and sustainability in the engineering field somehow depend on the right attributes being possessed in facing challenging and competitive demands in this profession.

The leadership role of engineers shows that the graduates produced are capable of deliberating and managing people, resources, technology, and the environment in creating a better lifestyle for society. With a strong sense of leadership instilled, it is an advantage for engineers to move the organisation they lead into greater heights and success. Not only are they knowledgeable in their area of expertise but they can also lead members of staff within groups and organisations in the most technical and critical managerial decisions. It is therefore understandable why leadership is an important trait for engineers to have [23]. Leadership is an ability to lead an organisation, especially when the scope of the job widens, requiring the capacity to handle major projects. Not only are engineers technically trained but they are also capable of managing big organisations, which is the ultimate goal in producing highly capable engineers.

During PEO revision, it is highly recommended that an internal committee within the institutions be formed in order to add these values in accordance with the acceptance of this analysis. The various views taken from stakeholders should be a priority as well as in understanding the current needs with a preferred choice of popular demand. The oversight of these attributes will give a negative impact on the institutions, which will not only downgrade the ranking and quality of their future engineers but will also lower the perceptions of graduates from those particular universities. The study has proven that the majority of the universities in Malaysia and other well-known universities worldwide do accept that the three attributes be made compulsory for engineering graduates to attain for the sake of their future careers and in their ability to work professionally.

*7.2. Evaluation of PEO Achievement and Continuous Quality Improvement*

A PEO survey is performed to evaluate the effectiveness of the academic management system in producing highly competitive graduates. The achievement of the alumni (three years upon graduation) represents the success of the programmes which embodied their graduates with the targeted attributes. The evaluation of PEO achievements is carried out with the following methods:

(a)    Stakeholders' (alumni) survey;
(b)    Prospective employers' survey;
(c)    Evaluation of the programme outcomes (POs) of all courses.

Our institution is deploying frameworks to monitor the progress of PEO achievement on engineering graduates. The first approach is using the stakeholders' survey as a direct measurement of PEO achievement. Meanwhile, a prospective employers' survey is carried out as part of the industrial training course, and PEO achievement is inferred from this survey. This constitutes an indirect measure as this survey is on students who have yet to undergo their fourth year of the programme. The final method, using an evaluation of the POs, is considered as another indirect measurement as the PO achievement scores are mapped to infer the PEO achievement score. The indirect measures can thus be used to project the PEO achievement of students even before they graduate. This is used in the improvement of PEO achievement. The overall description of the process of evaluating PEO achievement is as illustrated in Figure 19.

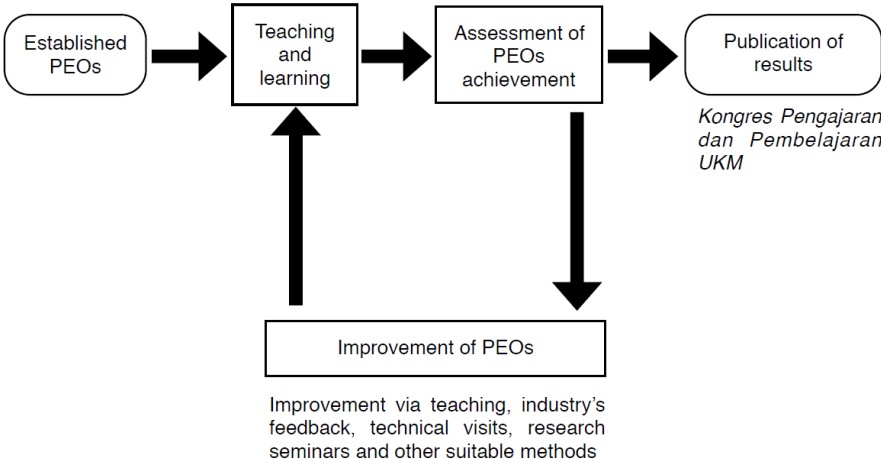

**Figure 19.** Flowchart of the process of assessing PEO achievement.

To evaluate the achievement of the PEOs by the graduates, a comprehensive online survey has been developed and can be accessed at http://www.ukm.my/peo (accessed on 10 August 2021). The survey is comprised of 21 questions, each of which is directly mapped to a specific PEO. The answer obtained for each question can provide a quantitative measure on the achievement of a PEO by the respondent. A snapshot of the online survey form is shown in Figure 20, referring to a sample of the results that are used to demonstrate the performance and achievement from PEO surveys.

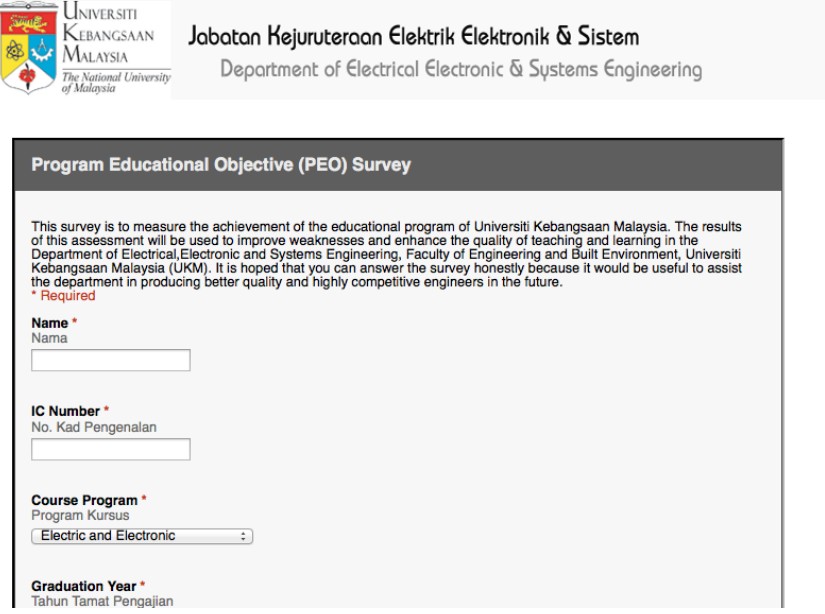

**Figure 20.** Online PEO survey.

The prospective employers' survey is annually conducted in conjunction with the industrial training course, during which comprehensive questionnaires are distributed to the employers. In addition, visits conducted by our academicians to the employers at the training sites also provide first-hand feedback on how the employers rate our current students. The employers' feedback also carries a certain weightage on the grades of the students enrolled in the industrial training course.

The department is committed to continuously improve on meeting and delivering all of the PEOs. The CQI initiatives undertaken by the department are two-pronged—to improve the results, and to improve the methods of measuring the results. The department

employs both direct (alumni survey) and indirect (industrial training survey and PO-PEO mapping) methods of measuring PEO achievement. Hence, both methods of measurement are used to evaluate and improve PEO achievement. The procedure of evaluation and improvement of PEO achievement is represented by Figure 21.

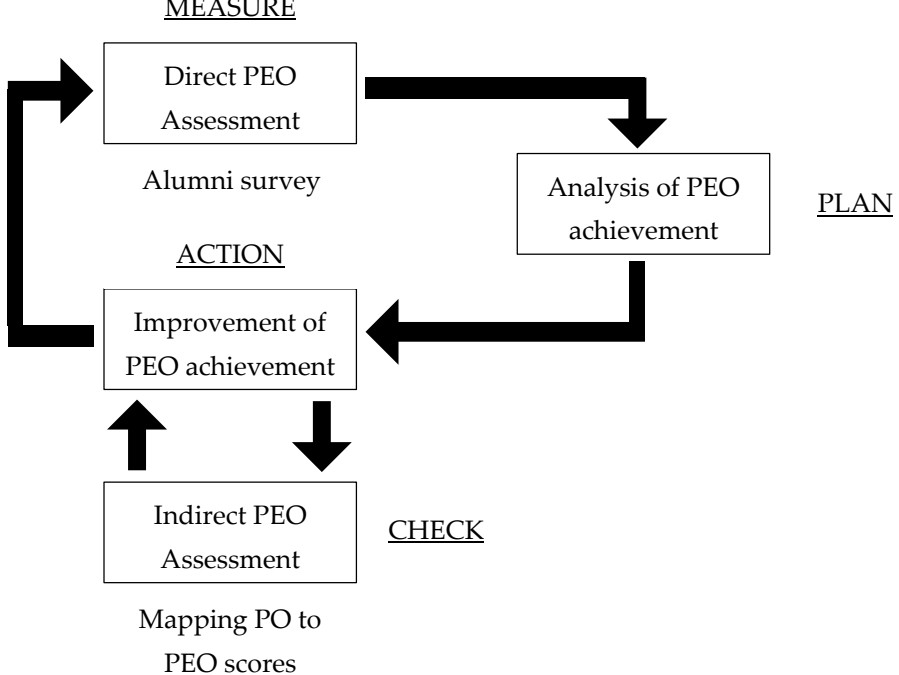

**Figure 21.** Flowchart of the process of assessment and improvement of PEO achievement.

The cycle of measuring and improvement of PEO achievement can thus be summarised in three steps:

i.   Direct PEO assessment (MEASURE);
ii.  Analysis of PEO achievement (PLAN);
iii. Improvement of PEO achievement (ACTION).

Direct PEO assessment and analysis have been presented. For those who graduated between 3 and 5 years ago, all target achievements (Tas) for all PEOs have been met in this assessment. The action plan, in this case, is thus to maintain and possibly improve upon these achievements. The actions that the department takes to maintain and improve PEO achievements can mainly be divided into two categories:

i.  Efforts on current students;
ii. Efforts on graduates.

In the first category, as the actions (ACTION) are performed during the undergraduate education of the students while the direct PEO assessment is performed between three and five years after they have graduated, indirect PEO assessment is performed during their studies to serve as a checking mechanism (CHECK). These indirect measurements have been described where PO scores are mapped to PEO scores; it can be seen on average that there is a slight overall increase in PEO achievement throughout the undergraduate programme at the department, and this gives some indication of improvement in PEO achievement, which would continue after the students graduate. In the second category, i.e., efforts on graduates, the checking is performed through direct measurement (MEASURE), as mentioned above. Based on the results obtained, the department initiates a series of initiatives aimed towards continuous improvement of the PEOs and overall effectiveness of the programmes. Our results prove to be significant for improving the standard of engineering graduates as tools for continuous improvement.

The observed data is plotted on a graph that shows the overall alumni achievement (2017 to earlier) and target alumni (5 years of graduation and earlier). This information is shown in Figure 22 for both alumni data. Based on the performance target (Table 10), some attributes were achieved brilliantly while others were modestly achieved and require improvement. Achievement analysis shows two attributes that are not achieved—i.e., ethics, and creativity and innovation. Based on this information, the faculty undertakes many initiatives to ensure that these attributes are achieved excellently as academic seminars, academic plagiarism, and student engagement in competitions and others. In addition, the emphasis is also given in the relevant course. The student portfolio can be used to record the personnel achievement of student [24,25]. This initiative is beneficial to our students, especially during the interview process, as shows evidence of their competency.

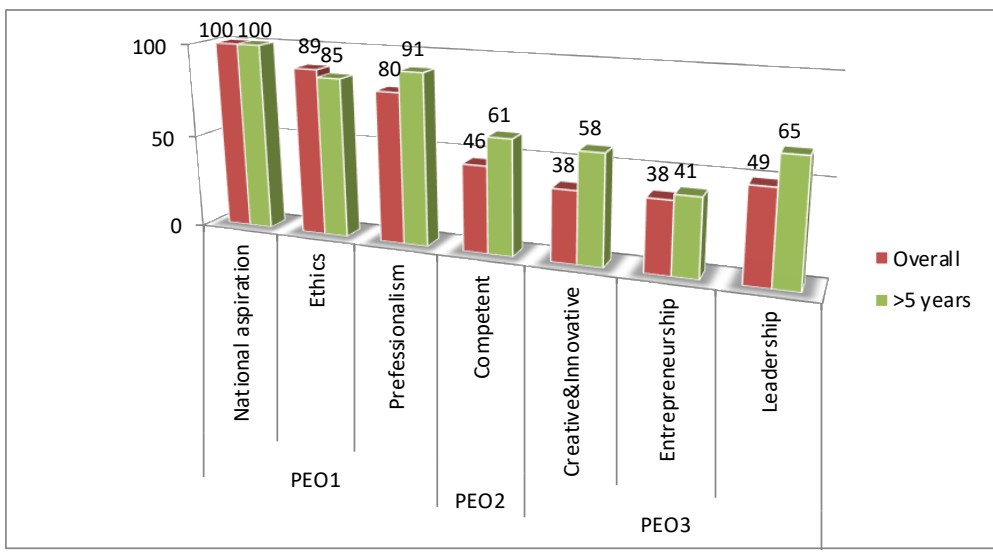

**Figure 22.** The difference in attainment of PEO attributes for the entire cohort and cohort graduated after 5 years.

**Table 10.** The target of achievement for every attribute.

| Attributes | National Aspiration | Ethics | Professionalism | Competent | Creative and Innovative | Entrepreneurship | Leadership |
|---|---|---|---|---|---|---|---|
| Target Achievement | 100 | 100% | 80% | 50% | 60% | 30% | 60% |

## 8. Conclusions

An assessment and review of different PEOs among Malaysian universities is one way to recognise the skills and attributes needed by electrical and electronic engineering graduates. The majority of EE engineering courses have at least three main domains identified throughout this research on public, private, or foreign branch university in Malaysia. Firstly, these electrical and electronic engineers are required to be professionally (PEO03) qualified with in-depth knowledge in their respected fields. Secondly, employers need talented engineers with leadership (PEO08) traits who can bring a significant impact and changes to their organisations. Thirdly, engineers are seen by most to be proficient in technical matters. It is therefore important for them to have high ethical principles and excellent professionalism (PEO07) added to their professional background. All three qualities are attributes capable of building the nation. Though the study is inconclusive, any changes in employability trends might in the future require new domains for EE engineers. The emergence of entrepreneurship is an example of how a multidisciplinary business platform might change engineering graduates' perceptions. The following phase of PEO development might see more features of current attributes appearing in the future. As the latest trend suggests, we are moving towards the fourth generation of the industrial

revolution (IR4), which co-relates to the needs of capturing new knowledge, skills, and attributes among engineering graduates globally. Our study is important to ensure the engineering programme is relevant, sustainable, and meeting the current industry demand.

**Author Contributions:** Conceptualisation, A.R.M.Y.; supervision, M.S.A.-R. and I.H.K.; project administration, A.W.M.; validation, I.-S.H. and A.K.A.M.I.; review and editing, J.A.R.; resources, M.J.M.N. All authors have read and agreed to the published version of the manuscript.

**Funding:** This research was funded by Universiti Kebangsaan Malaysia through Research-University grant (UKM-GUP-TMK-07-02-108).

**Institutional Review Board Statement:** Not applicable.

**Informed Consent Statement:** Not applicable.

**Data Availability Statement:** Not applicable.

**Acknowledgments:** This research was conducted in the Broadband, Network & Security Laboratory, Universiti Kebangsaan Malaysia (UKM).

**Conflicts of Interest:** The authors declared no potential conflict of interest with respect to the research, authorship, and/or publication of this article.

## Abbreviations

| | |
|---|---|
| UM | University of Malaya |
| UKM | National Universiti of Malaysia |
| USM | Universiti of Science, Malaysia |
| UTM | University of Technology, Malaysia |
| UPM | Universiti Putra Malaysia |
| UTeM | Technical University of Malaysia Malacca |
| UTHM | Tun Hussein Onn University of Malaysia |
| UMP | Universiti Malasya Pahang |
| UNiMAP | Universiti Malaysia Perlis |
| UPNM | National Defence University of Malaysia |
| UNITEN | Universiti Tenaga Nasional |
| MMU | Multimedia University |
| UNISEL | University of Selangor |
| UiTM | MARA University of Technolog |
| UTP | PETRONAS University of Technology |
| UCSI | UCSI University |
| UTAR | Tunku Abdul Rahman University |

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
