# Peer review of "A Global Program-Educational-Objectives Comparative Study for Malaysian Electrical and Electronic Engineering Graduates"

_sustainability, doi:10.3390/su14031280_

Round 1

Reviewer 1 Report

A Global Program Educational Objectives Comparative Study for Malaysia Electrical & Electronics Engineering Graduates. The main objective of this study is to determine which attributes must be 57 developed for secure employability in a currently demanded market. The result can be 58 used to evaluate the academic management system by evaluating the foremost 59 attributes for engineering graduates.

Overview of work: There is originality in this scientific work because the study is carried out in a developing country, with an efficient methodology and is a contribution to knowledge. The method used can be inspiring for other analyzes in equally developing countries. This paper contributes to the structuring of a now-tested methodology to set priorities in terms of knowledge, skills and competencies needed by undergraduate students of electrical and electronics engineering students. The literature used as reference is updated. The work is technically correct, the language is clear and explicit. Problem characterization in relation to the state of the art: There is an alignment between problem and objectives. The authors establish a dialogue with the various bibliographic references, conducting an efficient and effective reasoning that allows the reader to understand that the identified gap coincides with the paper's theme. The research method has its foundations in the literature, and is adjusted to the proposed theme. Bibliographic References: There is Relevance and Relevance. Scope and organization (current and classic; theoretical, specific and complementary), Use of bibliography published in journals with a high impact factor.

There is, however, an absence of description on how the literature review was carried out: which keywords, what are the steps and filter criteria.

Author Response

Thank you for the positive comment. We are appreciated it. The literature review has been explained explicitly on how it is carried out. The keyword has been inserted to highlight the key point or subject matter of the study. The element of sustainability has been highlighted in key word list as well as in the text.

Keywords: Accreditation; Programme educational objectives (PEO); graduates attributes; institutes of higher; learning (IHL); Electrical & Electronics (EE); sustainable engineering program

Reviewer 2 Report

  1. Need to improve as suggested comments marked in yellow in the manuscript.

Author Response

Thank you very much for the beneficial input to improve the quality of the article. We really appreciated to all the comments given.

Reviewer 3 Report

Please see attached

Author Response

First of all, thank you very much for the comments. We are really appreciated to the reviewer’s input.

All comments that have been raised by reviewer are addressed completed. The changes are highlighted with yellow colour in the main text.

Round 2

Reviewer 3 Report

OK. 

I still think that the authors should consider a journal that addresses engineering education as that readership would appreciate the content of the paper.